# Research on New Whitening and Water-Saving Technology Based on Industrial Equipment

**Yufei Chai, Weiting Jiang \* and Xin Zheng**

College of Energy and Mechanical Engineering, Shanghai University of Electric Power, Shanghai 200090, China; yufei970217@163.com (Y.C.); zxsuep@163.com (X.Z.)
**\*** Correspondence: wtjiang@shiep.edu.cn; Tel.: +86-18918302501

**Abstract:** Energy conservation and consumption reduction have always been the goals pursued by the power industry. Based on these goals, this study explored a new type of whitening and water-saving technology for industrial equipment. Unlike existing direct heating and condensation heating technologies, the main innovation of this work lies in not changing the saturated wet flue gas before it is discharged into the atmospheric environment. A series of experiments were conducted on electrode plates with different wind speeds, supersaturation levels, and porosities using three principles, namely droplet electrostatic adsorption, ionic-wind-enhanced condensation, and droplet dipole deflection, through the construction of a parameter-adjustable and -controllable enthalpy and humidity chamber and a pilot development platform with a wind volume of 10,000 $m^3$/h. In addition, the collection efficiency was calculated using thermodynamic laws. The results showed that, under the working conditions of white mist supersaturation of 5.77 g/kg, a hole opening rate of 70%, and a wind speed of 3 m/s, the water collection efficiency was the highest—close to 60%—verifying the feasibility of this technology. This technology not only eliminates white smoke but also saves water resources and has certain economic benefits, providing support for the development of industrial equipment for smoke removal in the future.

**Keywords:** electrostatic defogging; whitening; water conservation; industrial equipment

## 1. Introduction

As is well known, the formation of "white smoke" [1] is mainly due to the presence of a certain amount of water vapor in the ambient air, and its saturated moisture content is related to the temperature and pressure. The higher the temperature, the greater the saturated moisture content, and the stronger the ability to accommodate gaseous water. When saturated wet flue gas after wet desulfurization [2,3] is discharged into the atmospheric environment, due to the low temperature of the atmospheric environment, its vapor pressure is higher than the saturated vapor pressure of the atmospheric environment. The ability to hold gaseous water in the air is reduced, and the excess water vapor in the saturated wet flue gas condenses. Water vapor is the main component of flue gas, with a lower sulfide content. When high-temperature smoke, including sulfides, is discharged into the atmosphere, water vapor will be cooled and condensed into droplets containing sulfuric acid, which will precipitate into white mist in the form of droplets, known as "white smoke".

Concerning the whitening and water saving of industrial equipment, the main focus is on cooling towers. For the electrostatic defogging and water collection processes of the cooling tower studied in this article, a specially designed electrostatic defogging device is added at the outlet of the cooling tower. For the purpose of protecting the environment and saving resources, the whitening of industrial equipment is of great significance. Firstly, although the main component of the "white smoke" emitted by chimneys is the white mist that condenses when water vapor is cold, not true smoke and dust, it can reflect and refract

light to form different grayscales, which are not completely transparent, and is closely related to the intensification of visual pollution in cities. Secondly, from analyzing the generation mechanism of "white smoke", the flue gas after wet desulfurization is basically a saturated wet flue gas. The high-temperature flue gas entering the desulfurization tower absorbs a large amount of water vapor during the desulfurization slurry spraying process, and the flue gas contains a large amount of water vapor. The "white smoke" formed when discharged into the atmospheric environment is actually the process's water consumption. Taking a 600 MW unit as an example, where the flue gas volume is approximately $4 \times 10^6$ $Nm^3$/h, after wet desulfurization, the temperature of the flue gas entering the chimney is calculated at 50 °C, resulting in a loss of water consumption of about 410 T/h. In the northwest region of China, the annual water consumption cost of this unit is about CNY 16 million. It can be seen that the water content of saturated wet flue gas directly discharged from industrial equipment is very large, and the waste of water resources is extremely serious. If this water consumption can be recovered, it will have great economic benefits. Thirdly, the "white smoke" emitted by industrial equipment in power plants contains residual sulfides, which can cause corrosive damage to buildings and the surrounding environment. Condensate can absorb most of the solid particles, sulfates, nitrates, and other pollutants [4,5] in the smoke, reducing the emission of pollutants in the atmospheric environment and achieving the goal of environmental protection.

At present, research on flue gas whitening technology mainly focuses on MGGH technology, flue gas condensation dehumidification technology, solid dehumidification technology, etc. However, there is relatively little research on cyclone dehumidification and membrane dehumidification technologies [6–10], and these technologies are not yet mature. In 1944, Gaugler [11] invented heat pipes, and subsequently heat pipe technology was continuously studied, forming a relatively complete set of heat management theories. Li et al. [9] combined MGGH technology with traditional electrostatic precipitators, using high-ash coal as fuel, to test the emission characteristics and removal efficiency of particulate matter in a modified dust collector. The results showed that the total removal rate of particulate matter could reach 99.6%, and the removal rate of particulate matter increased with the increase in particle size and the decrease in inlet flue gas temperature. Zhang et al. [12] studied the low-temperature corrosion problem of MGGH systems and analyzed the superiority of enamel heat exchange tubes in terms of low-temperature corrosion resistance and enhanced heat transfer. Chen et al. [13] utilized condensing boilers to recover low-grade waste heat from flue gas and studied the economic and technological feasibility of condensing boilers using a large regional heating system as an example. Nipun Goel [14] applied Matlab to simulate the effectiveness of flue gas moisture recovery in five different arrangements of condensing heat exchangers. Terhan [15] conducted a study on the recovery and utilization of flue gas waste heat in a 60 MW gas central heating system of a certain university. The results showed that the flue gas condenser, which is made of a 316 stainless steel material with a surface area of 80 $m^2$, can condense flue gas to below 40 °C and has high economic efficiency. Yuzhong Li et al. [16] proposed the FEC-HP method, which uses a spray tower to deeply cool the flue gas to recover condensate and waste heat, and then completes the regeneration of the coolant through flash condensation and heat pump devices. Zhu et al. [17] designed a system combining a contact heat exchanger and an absorption heat pump on a gas boiler, and the results showed that after the modification of the system, the heating rate could be increased by 12%. Li et al. [18] used a combination of direct combustion absorption heat pump and direct contact heat exchanger technologies to retrofit a 29 MW gas boiler, and tested the entire heating season. The results showed that the contact heat exchanger could reduce the flue gas temperature to 27.4 °C, achieve an average heat recovery of 2.67 MW, and improve the heating efficiency by 11.54%. In addition, this technology has good effects in energy conservation and emission reduction and is feasible in large-scale applications. Li et al. [19] proposed a new method for finding ideal hygroscopic solutions using the NRTL equation and applied it to a mixed solution of calcium chloride and lithium chloride to calculate the

mixing ratio with a good dehumidification effect under certain conditions. Lars Westerlund et al. [20] utilized an open absorption system to recover waste heat and remove particulate matter from biofuel boilers. The recovered waste heat was used to heat the return water of the heating network, and the solution regeneration heat source came from the boiler flue gas. The experimental results showed that compared with traditional systems, the particulate matter in the flue gas was reduced by 33 to 44%. When the moisture content of biofuels is high, the heating capacity of the unit can be increased by nearly 40%. In terms of performance research on dehumidification solutions, H.T. Chua [21] proposed a set of fitting formulas for parameters such as the specific enthalpy and surface vapor pressure of a lithium bromide aqueous solution. The formulas are applicable to temperatures ranging from 0 to 190 °C and concentrations ranging from 0 to 75%. Conde [22] summarized a large amount of experimental data and provided fitting formulas for the crystallization curve, surface vapor pressure, density, surface tension, thermal conductivity, and specific enthalpy of two dehumidification solutions—lithium chloride and calcium chloride—which can be used for the comparison and theoretical calculation of dehumidification solutions. Mechanical ventilation wet cooling towers (MCTs) control the generation of plumes and save water resources [23,24].

The existing industrial equipment whitening and water-saving technologies, whether they are direct heating or condensation heating technologies, can reduce the relative humidity of wet flue gases. During the process of wet flue gas transforming into ambient air, it will not become saturated or form wet smoke plumes, ensuring that there is no "white smoke" when discharged into the atmosphere. However, these technologies rely on large heat exchange equipment, as well as cold and heat sources, which inevitably means that they have high investment and operating costs. This article introduces the concept of industrial equipment whitening and water-saving technology. By summarizing the previous ideas and innovating them, the saturated wet flue gas is no longer changed before being discharged into the atmospheric environment. When the saturated wet flue gas is discharged into the atmospheric environment, the atmospheric environment is used as a natural cold source to generate white mist. Then, the white mist is collected and recycled using high-voltage electrostatic devices. While the white mist is being captured, whitening is also achieved. At the same time, water recycling is completed. Because this technology uses nature as a cold source without the need for additional cold sources, it greatly reduces equipment investment and operating costs. At the same time, "mist" can be recycled in the form of water flow, which can supplement the water used in the wet desulfurization process and has water-saving effects.

## 2. Technical Principles

### 2.1. Electrostatic Adsorption of Fog Droplets

Inspired by the principles of electrostatic dust removal technology [9,25], when a small liquid droplet is placed near an electrode with positive or negative charges, the electrode generates a strong electric field, and the droplet is subjected to the electric field force. When the droplet is small enough, the electric field force is sufficient to overcome the droplet's own tension and move it towards the electrode, forming the principle of electrostatic dust removal technology.

Take the application of a DC high voltage to a group of electrode pairs with a significant difference in the curvature radius as an example. When the electric field intensity between the electrode pairs reaches a certain value, the air molecules between the electrode pairs will be ionized, generating a large amount of positive and negative space charges. Under the action of the electrostatic field, the positive and negative charges will move towards the two electrodes separately, as shown in Figure 1.

When the white mist flows through the above electric field, the droplets in the white mist will capture charges and become charged, and the charged droplets will also undergo directional movement under the action of the electrostatic field. Assuming that an electrode with a small radius of curvature applies a negative high voltage, and an electrode with a

large radius of curvature is grounded at the low-voltage end, the positive charge in the electric field will flow to the electrode with a small radius of curvature, while the negative charge will flow to the electrode with a large radius of curvature, forming a current circuit. After the droplets capture charge, they are separated into two electrodes under the action of an electrostatic field. The droplets continuously adhere to the surfaces of the two electrodes and are collected by gravity in the form of water flow.

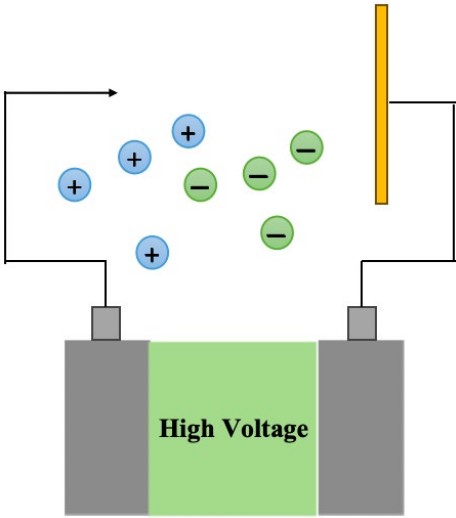

**Figure 1.** Schematic diagram of electrostatic adsorption of fog droplets.

### 2.2. Ionic-Wind-Enhanced Condensation

The electrostatic whitening technology explored in this article electrostatically captures the white mist generated by saturated wet flue gas discharged into the atmospheric environment through industrial equipment. Therefore, the better the condensation effect of ambient air on the saturated wet flue gas discharged into the atmospheric environment, that is, the higher the concentration of white mist generated, the better the whitening effect of this technology.

In the process of electrostatic adsorption of droplets as explained in Section 2.1, air ionization generates a large amount of positive and negative charges that continuously move in the electrostatic field. The continuous directional movement of the charges will generate ionic winds perpendicular to the flow direction of the wet air main body.

Analysis of the electric field spatial flow field using the particle laser imaging velocimetry (PIV) method shows that the ionic wind generated by discharge can effectively improve the turbulence intensity of the flow field, and at the same time, many negative pressure zones will be formed near the edge of the electric field. The increase in turbulence intensity in the flow field is conducive to the heat and mass transfer between the saturated wet flue gas and ambient air. The negative pressure zone at the edge of the electric field can further entrain ambient air, which is conducive to the condensation of saturated wet air into mist.

### 2.3. Droplet Dipole Deflection

The two H-O bonds of water molecules do not coincide at an angle of 104°5′, resulting in a mismatch between the centers of gravity of the positive and negative charges of water molecules. Due to the atomic mass of oxygen atoms being greater than that of hydrogen atoms, oxygen atoms have a stronger ability to attract charges, causing the four electrons (Represented by black dots in Figure 2) that have not formed H-O bonds to lean towards oxygen atoms. This causes the water molecule to exhibit local electronegativity, and the negative charges carried by oxygen atoms and the positive charges carried by hydrogen atoms form a pair of electric dipoles.

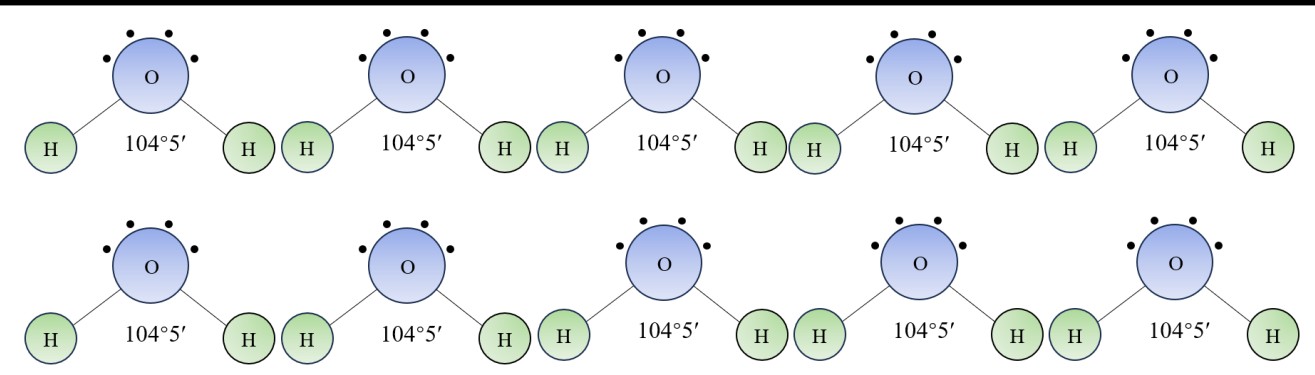

**Figure 2.** Droplet dipole deflection.

The droplets contain a large number of water molecules, which are linked by hydrogen bonds. Without the action of external electric and magnetic fields, the arrangement of water molecules in the droplets is disorderly. When a droplet enters the electric field space, the water molecules in the droplet undergo directional deflection. As shown in Figure 2, the black line represents a positive potential, while the red line represents a negative potential. The water molecules in the droplets undergo a regular arrangement, with oxygen atoms driven towards a positive potential and hydrogen atoms driven towards a negative potential. When the electric field between the positive and negative potentials is an uneven electric field, due to the large mass of the oxygen atom, the gravitational force of the electrostatic field is greater than that of the hydrogen atom. Therefore, the oxygen atom carries the hydrogen atom and moves towards the positive potential electrode, causing the fog droplets to undergo directional deflection in the extremely uneven electrostatic field, moving to the surface of the positive potential electrode to form a water flow for recycling.

## 3. Experimental Part

### 3.1. Experimental Equipment

Small and medium-sized R&D platforms can be built, which should have gas distribution systems, wind power systems, heating systems, water vapor generation systems, high-voltage electrical systems, and electrical and thermal detection systems. Simulation tests can be conducted based on actual environmental conditions.

The small- and medium-scale experimental system used in this work included an air distribution device, a water vapor generation device, a high-voltage power supply, an electrostatic water collector, a weighing device, and thermal instruments and meters. The pilot research and development platform is shown in Figure 3. There was a total of two enthalpy and humidity chambers, divided into the outdoor side and the indoor side. The outdoor side was a mixing chamber for water vapor and ambient air, and the indoor side was an electrostatic defogging experimental chamber. The volume of a single room was 125 m$^3$, and the temperature range was adjustable and controllable from 0 °C to 45 °C. The relative humidity range was adjustable and controllable from 20% to 100%. The enthalpy and humidity chamber was able to simulate different environmental temperatures and humidities, and the temperature and humidity of the environmental chamber could be adjusted and controlled according to actual environmental conditions. Outside of the room, steam was mixed with the ambient air of a fan through a pipeline to generate saturated moist air at a certain temperature, which entered the indoor side and was used to simulate electrostatic mist removal experiments under different actual temperature conditions.

### 3.2. Experiment Approach

Based on the three principles of the electrostatic adsorption of fog droplets, enhanced condensation by ionic winds, and deflection of fog droplet dipoles, saturated wet air at 50 °C was simulated, and a cylindrical mist eliminator was designed to eliminate the

white mist generated. Data processing and economic analysis were conducted to verify the feasibility of this technology.

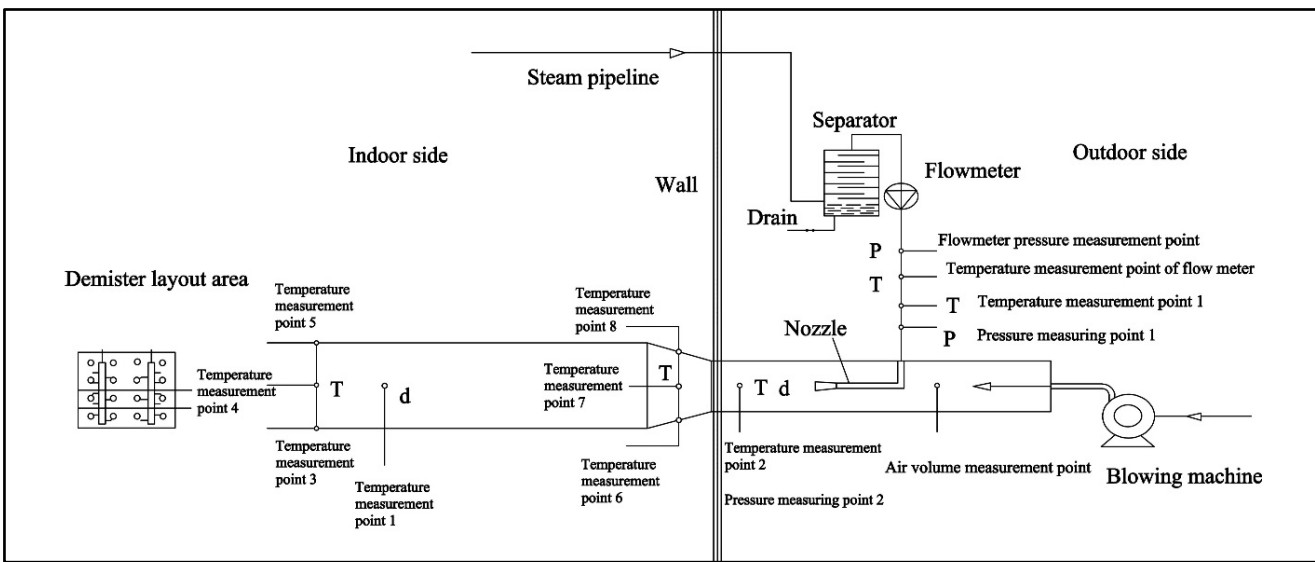

**Figure 3.** A pilot research and development platform with adjustable and controllable parameters for an enthalpy and humidity chamber and 10,000 m$^3$/h air volume.

(1) Outdoor side

The outdoor side served as the operating environment for the fan, which could be stably controlled under constant temperature conditions of dry bulb 20 °C and wet bulb 15 °C, with a fluctuation range of less than 1 °C. After opening the steam valve, the water vapor in the pipeline first entered the gas–liquid separator for gas–liquid separation. The liquid water was discharged through the drainage valve, and the gaseous water vapor was discharged from the gas–liquid separator where it first flowed through the water vapor flow meter. Then, it entered the air duct and mixed with the ambient air of the fan.

Along the direction of steam flow, the measurement points included the following, in order: a steam flow meter, the pressure measurement point of the flow meter, the temperature measurement point of the flow meter, remote temperature measurement point 1, and mechanical pressure gauge 1 (changed to a communicable pressure gauge). Along the direction of environmental air flow, there were the air volume test point (pitot tube micro pressure gauge), temperature test point 2, and humidity test point 2.

(2) Indoor side

After passing through the wall and then through the circular area, the mixed moist air first flowed through a 350 mm × 500 mm × 2500 mm square tube and was then fed into the electric mist eliminator and finally discharged into the indoor space. For the temperature control of the indoor side, when the temperature was set to dry bulb 5 °C, after actual testing, under the conditions of a 5.19 m/s wind speed and 30 kg/h water vapor flow rate, the temperature range in the morning was 5~6.5 °C, and in the evening, the temperature increased to 11 °C. When the temperature was set to dry bulb 15 °C, actual testing showed that under the conditions of a 9.58 m/s wind speed and 260 kg/h water vapor flow rate, the temperature of four compressors could be stably controlled during operation, and the temperature of three compressors could be increased to 17 °C during operation.

The outlet of the square tube was not directly connected to the inlet of the electric mist eliminator, and a neutral position was specially set to prevent the condensed water inside the square tube from entering the electric mist eliminator. The outlet of the square tube and the outlet of the electric mist eliminator both had a water collection tray, and the condensed water in the square tube and the collected water in the mist eliminator were collected and weighed independently. Along the direction of wet air flow, the measurement points were

as follows: temperature measurement points 6, 7, and 8; humidity measurement point 1; and temperature measurement points 3, 4, and 5.

### 3.3. Method of Calculation

3.3.1. Theoretical Calculation of Supersaturation

The known parameters are as follows: air volume flow rate $V$, outdoor wet air dry bulb temperature $T_{dry,out}$, and wet bulb temperature $T_{wet,out}$. Some symbols in the equations are explained in Table 1.

**Table 1.** Symbol description.

| Symbol | Definition | Unit |
|--------|------------|------|
| $d$ | Moisture content | g/kg |
| $T$ | Temperature | °C |
| $V$ | Volumetric flow rate | m$^3$/h |
| $\rho$ | Density | kg/m$^3$ |
| $\varphi$ | Relative humidity | % |
| $\eta$ | Efficiency | % |
| $h$ | Specific enthalpy | kJ/kg |
| $m$ | Mass flow | kg/s |
| $P$ | Pressure | MPa |
| $in$ | Indoor side humid air | - |
| $out$ | Outdoor side humid air | - |
| $0$ | Indoor environment | - |
| $dry$ | Dry bulb | - |
| $wet$ | Wet bulb | - |

Mass conservation equation:

$$m_{steam} + m_{dry}d_{out} = m_{super} + m_{dry}d_{in} \tag{1}$$

Energy conservation equation:

$$m_{steam}h_1 + m_{dry}h_{out} = m_{super}h_{super} + m_{dry}h_{in} \tag{2}$$

Dry air mass flow rate $m_{dry}$:

$$m_{dry} = \frac{V\rho_{out}}{(1 + d_{out})} \tag{3}$$

Outdoor wet air parameters $d_{out}, \rho_{out}, h_{out}$:

$$(\rho_{out}, d_{out}, h_{out}) = f\left(T_{dry,out}, T_{wet,out}\right) \tag{4}$$

Indoor side wet air dry bulb temperature $T_{dry,in}$:

$$T_{dry,in} = \frac{T_3 + T_4 + T_5}{3} \tag{5}$$

Indoor wet air parameters $d_{in}, \rho_{in}, h_{in}$:

$$(\rho_{in}, d_{in}, h_{in}) = f\left(T_{dry,in}, \varphi_1\right) \tag{6}$$

Specific enthalpy of steam $h_{steam}$:

$$h_{steam} = f(T_1, P_1) \tag{7}$$

Specific enthalpy of supersaturation $h_{super}$:

$$h_{super} = f\left(T_{dry,in}, P_0\right) \tag{8}$$

### 3.3.2. Demisting Efficiency

$$\eta = \frac{m_W}{m_{super} - m_{cond}} \times 100\% \tag{9}$$

$W$ is the amount of water harvested.

### 3.3.3. Electrical Power

$$P = UI \times \kappa \tag{10}$$

In the equation, $\kappa$ is the power factor, which is dimensionless and related to the proportion of power output from the power source.

$$\kappa = \frac{1}{1 + e^{-0.05(p-50)}} \tag{11}$$

$$p = \frac{UI}{P} \tag{12}$$

In the formula, $p$ is the proportion of power output from the power supply, and P is the rated power of the power supply.

## 4. Results and Discussion

Through small-scale theoretical verification experiments, it was found that the outlet of the mist eliminator could eliminate white smoke under conditions of 15~20 °C. A comparative analysis was conducted on the water collection and power consumption, as shown in Table 2. In an ideal state without considering energy consumption, equipment costs, etc., the absolute amount of collected water was observed after 30 min of water collection, with the highest economic efficiency of 692 kg/kW·h.

**Table 2.** Comparison and analysis of harvested water and electricity consumption.

| Absolute Amount of Water Received (kg) | Power (W) | Energy Consumption Output (kg/kW·h) |
|---|---|---|
| 0.012 | 0.018 | 121 |
| 0.015 | 0.0986 | 304 |
| 0.014 | 0.0564 | 496 |
| 0.013 | 0.0566 | 458 |
| 0.008 | 0.225 | 70 |
| 0.019 | 0.1416 | 268 |
| 0.031 | 0.0895 | 692 |
| 0.015 | 0.0663 | 452 |

By using the calculation formula in Section 3.3, the water recovery rates under different supersaturation amounts and pore opening rates can be obtained, as shown in Table 3.

Then, the experimental data can be plotted in a chart, as shown in Figure 4.

As can be seen from Figure 4 above, under the same supersaturation condition, the mist removal efficiency of the electric mist eliminator shows a trend of first increasing and then decreasing with the increase in the opening rate of the orifice plate, and the inflection point appears near the 70% opening rate. For orifice plates with the same opening rate, as the supersaturation gradually increases, the demisting efficiency of the electric demister also shows a decreasing trend. Among them, when the supersaturation amount is

5.77 g/kg, the defogging efficiency is the highest, and when the supersaturation amount is 16.2 g/kg, the defogging efficiency is the lowest.

**Table 3.** Water recovery rate under different supersaturation levels and pore opening rates.

| Supersaturation | | | | | | Porosity (%) | Water Collection Capacity (kg) | Water Collection Rate (%) |
|---|---|---|---|---|---|---|---|---|
| kg/h (Not Deducted) | kg/h (Deducted) | g/m³ (Not Deducted) | g/m³ (Deducted) | g/kg (Not Deducted) | g/kg (Deducted) | | | |
| 8.22 | 6.25 | 6.01 | 4.57 | 5.77 | 4.39 | 55 | 3.295 | 52.720 |
| | | | | | | 70 | 3.643 | 58.280 |
| | | | | | | 80 | 2.861 | 45.782 |
| | | | | | | 90 | 2.028 | 32.450 |
| 4.38 | 3.76 | 3.20 | 2.75 | 2.91 | 2.50 | 55 | 1.844 | 49.042 |
| | | | | | | 70 | 1.779 | 47.316 |
| | | | | | | 80 | 1.562 | 41.540 |
| | | | | | | 90 | 0.983 | 26.136 |
| 13.57 | 10.48 | 9.94 | 7.676 | 10.41 | 7.85 | 55 | 4.861 | 46.380 |
| | | | | | | 70 | 4.467 | 42.626 |
| | | | | | | 80 | 3.537 | 33.754 |
| | | | | | | 90 | 2.267 | 21.632 |
| 11.17 | 8.23 | 8.105 | 5.97 | 8.34 | 6.145 | 55 | 3.609 | 43.851 |
| | | | | | | 70 | 3.490 | 42.410 |
| | | | | | | 80 | 3.305 | 40.155 |
| | | | | | | 90 | 2.479 | 30.121 |
| 17.95 | 15.86 | 13.22 | 11.69 | 12.38 | 10.94 | 55 | 1.623 | 10.234 |
| | | | | | | 70 | 1.430 | 9.014 |
| | | | | | | 80 | 1.184 | 7.468 |
| | | | | | | 90 | 0.691 | 4.357 |
| 20.49 | 17.314 | 15.23 | 12.87 | 15.25 | 12.89 | 55 | 1.924 | 11.110 |
| | | | | | | 70 | 1.500 | 8.661 |
| | | | | | | 80 | 1.182 | 6.825 |
| | | | | | | 90 | 0.791 | 4.568 |
| 16.40 | 14.57 | 12.09 | 10.75 | 11.04 | 9.816 | 55 | 2.044 | 14.031 |
| | | | | | | 70 | 2.072 | 14.221 |
| | | | | | | 80 | 1.272 | 8.733 |
| | | | | | | 90 | 0.720 | 4.942 |
| 15.70 | 11.91 | 11.43 | 8.675 | 12.31 | 9.34 | 55 | 3.934 | 33.027 |
| | | | | | | 70 | 3.808 | 31.970 |
| | | | | | | 80 | 3.053 | 25.634 |
| | | | | | | 90 | 1.925 | 16.161 |
| 17.02 | 13.115 | 12.28 | 9.46 | 16.2 | 12.49 | 55 | 3.595 | 27.413 |
| | | | | | | 70 | 3.184 | 24.277 |
| | | | | | | 80 | 2.825 | 21.539 |
| | | | | | | 90 | 1.437 | 10.957 |
| 15.32 | 11.08 | 11.22 | 8.11 | 12.51 | 9.05 | 55 | 3.957 | 35.717 |
| | | | | | | 70 | 3.730 | 33.660 |
| | | | | | | 80 | 3.338 | 30.123 |
| | | | | | | 90 | 1.922 | 17.349 |
| 14.54 | 10.91 | 10.363 | 7.78 | 11.06 | 8.3 | 55 | 4.151 | 38.050 |
| | | | | | | 70 | 3.974 | 36.421 |
| | | | | | | 80 | 3.722 | 34.12 |
| | | | | | | 90 | 3.478 | 31.88 |

Not Deducted refers to the theoretical value calculated based on the conservation of mass and energy equations. Deducted refers to the calculated theoretical value minus the on-site condensation yield value.

Through the above research and development experiments, it is also possible to compare electrostatic water collection with traditional thermal water collection. The comparison results are shown in the following figures.

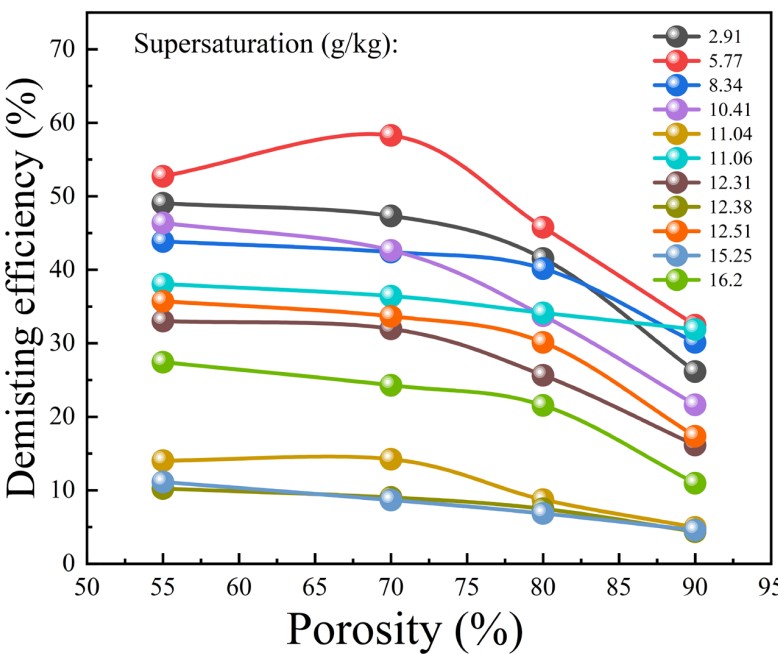

**Figure 4.** Water recovery rate under different supersaturation levels and pore opening rates.

The water collection test was conducted on the indoor side with a controlled temperature of 5 °C and a relative humidity of 80%, as shown in Figure 3. "Turn off the mist eliminator" represents the condensation of 35 °C saturated wet air under 5 °C environmental cooling conditions, which is the best water-saving condensation method in conventional thermal processes. The experimental results in Figure 5 indicate that when using the high-voltage electrostatic water collection technology, eight to nine times more water can be collected than when using the conventional condensation method, which means that the same amount of water is collected. The area of the high-voltage electrostatic water collection device only needs 1/8~1/9 of that of the conventional condensation method. The results of this experiment indicate that the high-voltage electrostatic water collection technology has a high water-saving rate and requires a small equipment area.

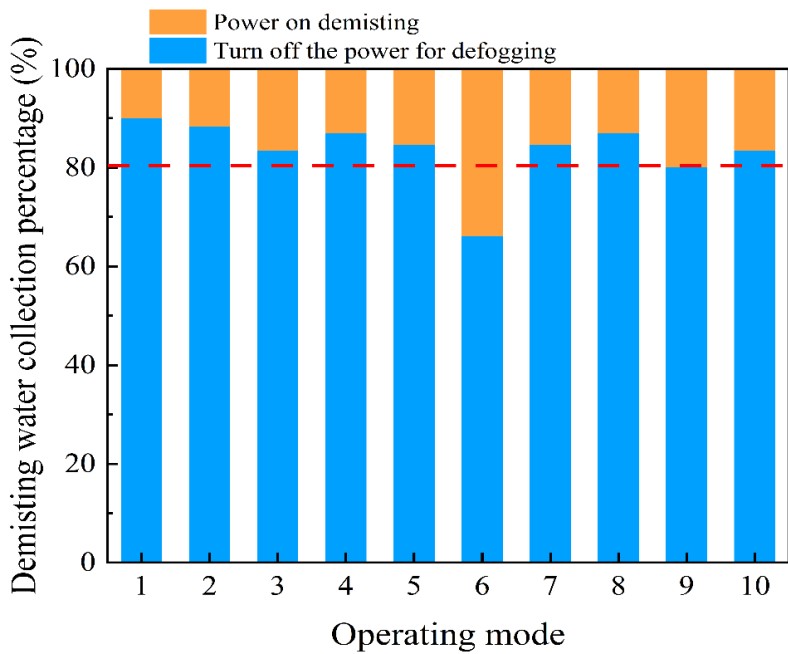

**Figure 5.** Comparison between electrostatic water collection and condensation water collection.

The only energy this technology consumes is electrical energy, and it is necessary to evaluate the economy of both the electrical energy consumption and water collection. As shown in Figure 6, among the 10 working conditions tested, the minimum economic efficiency is 407 kg of water collected at 1 Kw·h, the maximum economic efficiency is 1450 kg of water collected at 1 Kw·h, and the average economic efficiency is 670 kg of water collected at 1 Kw·h. In the northwest region of China, the grid electricity price is calculated at CNY 0.5, and the price of 1 ton of industrial water is calculated at CNY 5. Within the range of the results of this experiment, even if the smallest economic result is calculated, the consumption of 1 kW·h of electricity is calculated at CNY 0.5, but 407 kg of water can be recovered. Based on the price of water being CNY 2, the net value is CNY 1.5, indicating that the economy of using electrostatic water harvesting technology is extremely considerable.

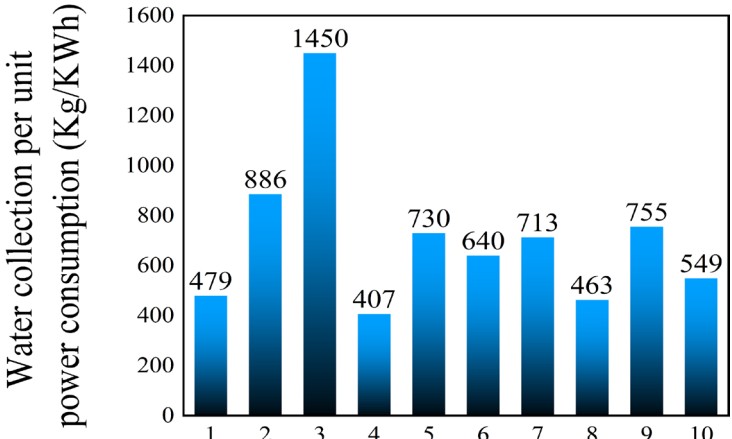

**Figure 6.** Water collection per unit power consumption.

In our water collection experiments with seven types of water collection electrode opening rates and four types of high-pressure electrodes, the distribution of the mist removal efficiency under various operating conditions was relatively concentrated, ranging from 50% to 70%, with an average mist removal efficiency of 59%. The mist removal efficiency under different pore rates and electrode structure conditions is shown in Figure 7.

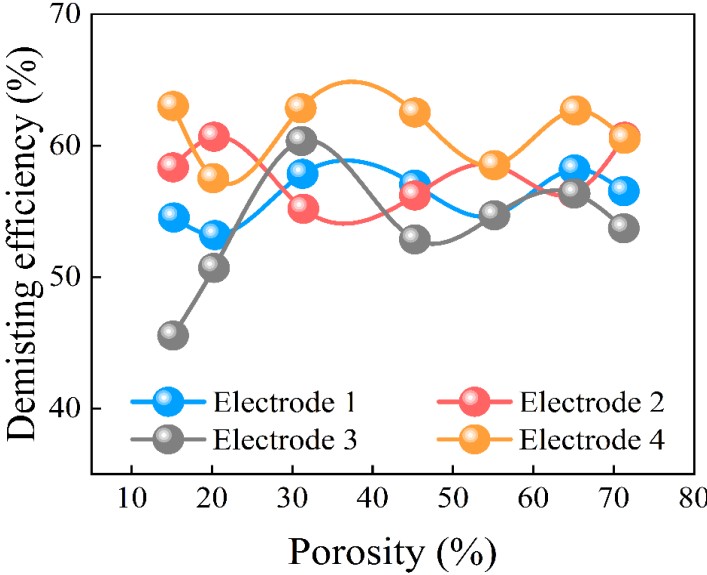

**Figure 7.** Demisting efficiency under different pore rates and electrode structure conditions.

Taking the electrostatic water collection project of a 300 t/h circulating water volume mechanical cooling tower as an example, the evaporation capacity is 4.5 t/h. Calculated based on the conventional exhaust temperature of 35 °C, the saturated moisture content corresponding to 35 °C is 36.55 g/kg. Calculated based on the annual average environmental temperature of 12 °C in northern China, the saturated moisture content corresponding to 12 °C is 8.72 g/kg, resulting in a condensation capacity of 3.426 t/h, with a water yield of 2 t/h.

If the unit operates for 8000 h throughout the year, the annual water yield is 16,000 tons. According to the average economic index of the pilot test, the annual electricity consumption is 23,880 kW·h, calculated as 670 kg of water per kWh of electricity. If the grid electricity price is calculated at CNY 0.5 and the industrial water price for 1 ton is calculated at CNY 5, the annual saved water cost is CNY 80,000, the electricity consumption is CNY 11,900, and the net value is CNY 68,000.

## 5. Conclusions

In summary, this study has successively verified the feasibility and water-saving economy of an electrostatic mist removal and whitening technology through small-scale and pilot experiments. The following are the main conclusions drawn through the experiments.

Through small-scale theoretical verification experiments, it was found that the outlet of the mist eliminator can eliminate white smoke under conditions of 15~20 °C. After 30 min of water collection, the highest economic efficiency of 692 kg/kW·h could be achieved, which verifies the theoretical feasibility of this technology.

Through pilot experiments in an environmental chamber, it was found that, under the same supersaturation condition, the mist removal efficiency of the electric mist eliminator showed a trend of first increasing and then decreasing with the increase in the opening rate of the orifice plate, and the inflection point appeared near the 70% opening rate. For orifice plates with the same opening rate, as the supersaturation gradually increased, the demisting efficiency of the electric demister also showed a decreasing trend. Among them, when the supersaturation amount was 5.77 g/kg, the defogging efficiency was the highest, and when the supersaturation amount was 16.2 g/kg, the defogging efficiency was the lowest.

By comparing electrostatic water collection with traditional thermal water collection, the results show that the amount of water collected using the high-voltage electrostatic water collection technology was 8–9 times more than that collected using the traditional condensation method. It can be inferred that the amount of water collected is the same. The area of the high-voltage electrostatic water collection device only needed 1/8 to 1/9 of that of the traditional condensation method. From this, it can be concluded that the electrostatic defogging water-saving device occupies a small area. The defogging efficiency is high, and after evaluating the economic benefits of this technology, it was found that the economic benefits are quite considerable.

At present, China's industrial energy consumption is enormous. The successful development of electrostatic defogging technology can be applied to the cooling towers of thermal power plants, which can greatly save industrial water and has good prospects for achieving sustainable and green industrial development. However, this technology may have drawbacks such as electrode contamination and difficulty in controlling the electrode spacing. Exploring how to break through technical barriers will be the main focus of future research.

To sum up, this technology has the advantages of a high water-saving efficiency, stable operation, safety and reliability, a single energy consumption form, low operating costs, a high energy output rate, significant economic benefits, modularity, and easy construction, installation, and maintenance, as well as being able to simultaneously remove multiple pollutants. However, there are also disadvantages, such as electrode contamination, difficulty in controlling electrode spacing, and potential voltage instability. Next, we will continue to study the scalability of this technology and contribute to its sustainable development.

**Author Contributions:** Formal analysis, X.Z.; Data curation, W.J.; Writing—original draft, Y.C. All authors have read and agreed to the published version of the manuscript.

**Funding:** This research received no external funding.

**Data Availability Statement:** Data are contained within the article.

**Conflicts of Interest:** The authors declare that they have no known competitive financial interests or personal relationships that may affect the work reported in this paper.

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
