# Peer review of "Research on New Whitening and Water-Saving Technology Based on Industrial Equipment"

_energies, doi:10.3390/en17051052_

Round 1
Reviewer 1 Report
Comments and Suggestions for Authors
This work focuses on the application and test of electrostatic water harvesting technology adopted in smoke removal. The topic is meaningful for the power engineering. The following points are suggested to improve the manuscript as:
1) The innovation of the study should be enhanced.
2) The quality of Figs. 3-4 and Tables 1-2 should be promoted.
3) In the last paragraph, the word "I" should be modified.
4) In the Sec. 3.2, the word "calculate" in some title can be deleted.
Comments on the Quality of English LanguageCheck the full-text and make sure the specialized vocabularies are appropriate.
Reviewer 2 Report
Comments and Suggestions for Authors
The present study discussed very important subject. The paper is well written and in the scope of the journal, however, some minor comments should be addressed before accept to publication.
1. Please check the format of equation.And the equations lack an explanation of the relevant symbols.
2. Please check the affiliation. There should be two different affiliations.
3. Line 306: It is mentioned in table 2 that deduction refers to the calculated theoretical value minus the on-site condensation yield value. How to determine yield value in the field?
4. Figure 4-7:The scale of these figures is a little too large, please adjust.
5. References should be numbered in order of appearance and indicated by a numeral or numerals in square brackets—e.g., [1] or [2,3], or [4-6]. Please check.
6. The main text mentions the impact of "white smoke" on urban visual pollution, so what is the effect of this new technology on reducing urban visual pollution? How does it reduce emissions of pollutants in the atmosphere?
7. Please check the format of reference to be more formal. For example,[2]Chen, Q.; Finney, K.; Li, H.; Zhang, X.; Zhou, J.; Sharifi, V.; Swithenbank, J. (2012). Condensing boiler applications in the process industry. Applied Energy. 2012,89(1), 30-36. https://doi.org/10.1016/j.apenergy.2010.11.020
8. The literature review is incomplete, since several relevant and recent contributions are missing, for example:DOI:10.1139/cgj-2021-0361.
Comments on the Quality of English LanguageMinor revision may be required.
Author Response
Please refer to the attachment for experts.

Reviewer 3 Report
Comments and Suggestions for Authors
In this paper, the logic is clear and the research method is rigorous. Using the principles of droplet electrostatic adsorption and dipole deflection, a new type of water-saving and emission-reduction technology for industrial equipment has been developed, and the feasibility of this technology under different working conditions has been verified. This paper has a reference function for the development of the green industry. The work is suitable for publication in the magazine. However, there are some problems in this paper, which should be slightly modified before publication.
1. Some content in the introduction is excessive and the logic is vague, it is recommended to adjust or delete some content;
2. The font spacing in Tables 1 and 2 is too large;
3. The content of the experiment is insufficient, it is suggested to supplement it;
4. At the end of the discussion, the application of the technology in real industry should be prospected.
Comments on the Quality of English LanguageMinor editing of English language required
Author Response

(The authors gave the same response as above.)

Reviewer 4 Report
Comments and Suggestions for Authors
Dear Authors,
Here are some comments on: Research on new whitening and water-saving technology based on industrial equipment
The abstract lacks specific details regarding the aim of the research, experimental setup and methodology. Ensuring clarity, specificity, and contextualization of the methodology, results, and implications would strengthen the abstract's contribution to the field of industrial environmental management.
Introduction
Line 32: delete “a”
1. It is recommended to explicitly state the research objectives and the specific focus of the study within the Introduction. This will help orient the reader and provide a clear framework for understanding the relevance and significance of the research.
2. To enhance readability and coherence, the Introduction should be restructured to ensure a logical flow of ideas. Grouping related information together and using clear transitions between paragraphs will improve the overall organization of the Introduction.
Lines 116-117: The statement lacks specificity regarding how the state of the flue gas is changed. It would be more scientifically sound to provide details on the specific processes or mechanisms involved in altering the state of the flue gas.
Lines 125 -126: It is true that lower temperatures can lead to condensation of water vapor, but the statement does not provide sufficient detail on how the "white mist" is generated using the atmospheric environment as a cold source. More information on the physical processes involved would enhance scientific soundness.
Lines 130-131: The concept of recycling "white smoke" in the form of water flow is not scientifically accurate. "White smoke" typically refers to a suspension of water droplets in air, not a substance that can be readily converted into a liquid flow. This statement requires clarification or revision for scientific accuracy.
2. Technical Principles
Lines 134-143: The assertion that droplets will adsorb solely due to static electricity needs clarification. While electrostatic forces can influence droplet behavior, it's essential to provide more detailed mechanisms and factors involved in droplet adsorption.
Lines 176- 196: The explanation of droplet deflection could benefit from additional clarity and references to established scientific principles. Clarify the role of electric fields in droplet movement and provide evidence supporting the described mechanisms.
3. Results and Discussion
Lines 297-298: This statement suggests that the economic efficiency is solely determined by the absolute amount of water collected, which is misleading. Economic efficiency should consider various factors such as energy consumption, equipment cost, maintenance, and operational considerations.
Lines 311-313: This statement implies a direct causal relationship between the mist removal efficiency and the opening rate of the orifice plate without providing sufficient evidence or explanation. The relationship between these factors may be more complex and requires further analysis and validation.
Lines 330-332: The statement may be true based on the experimental results presented, but it lacks specific quantitative data and comparative analysis to support the claim. Additionally, the text does not provide information on potential limitations or drawbacks of the high-voltage electrostatic water collection technology.
Lines 340-346: This statement presents economic analysis based on specific assumptions and calculations. However, the validity of these conclusions depends on the accuracy of the assumptions and the relevance of the economic context provided. Without proper justification and sensitivity analysis, the economic conclusions may not be robust.
When presenting economic evaluations in a scientific paper intended for an international audience, it's generally advisable to provide conversions to widely recognized currencies such as US dollars or Euros. This allows readers from different regions to better understand the economic implications of the research findings and facilitates comparison with similar studies conducted in other parts of the world.
Conclusions
The conclusions drawn seem to be overly optimistic and generalized without providing specific data or context. It's important to provide more detailed information and evidence to support the conclusions, such as specific numerical data, statistical analysis, or comparisons with existing technologies. Additionally, it's crucial to acknowledge any limitations or uncertainties in the research and discuss avenues for future research to address these gaps.
The references appear to be relevant to the topic of the scientific paper. Each reference cited seems to contribute to the discussion of various aspects related to flue gas treatment, heat recovery, water saving, and related technologies.

Round 2
Reviewer 4 Report
Comments and Suggestions for Authors
Dear Editors,
After evaluation of the revisions made by the authors and considering the feedback provided, I recommend accepting the manuscript for publication after minor revisions.
However, there are still some minor discrepancies that need to be addressed.
Sincerely,
Assoc.Prof.PhD. Eleonora Nistor
